# XBOUND: Exploring Capability Boundaries of Device-Control Agents at the State Level

## Abstract

Recent advancements in vision-language models have increased interest in Device-Control Agents (DC agents) for managing graphical user interfaces (GUIs). With the growing complexity and integration of such agents into various applications, effective evaluation methods have become crucial. The current evaluation method for DC agents primarily focuses on the instruction level, providing the current state (e.g., screenshots) and past execution history to determine actions for target instructions, helping identify potential execution failures. However, in GUI environments, a single state may contain multiple interactive widgets, each linked to different instructions, presenting an opportunity for diverse actions based on various instruction targets. Evaluating the agent's performance solely at the instruction level may overlook the broader context of these interactions. To capture a more comprehensive view of agent performance, we propose a new evaluation method, XBOUND, to evaluate the accuracy of instruction completion on a per-state basis. XBOUND provides a state-level evaluation framework, serving as a tool to assess agents' capabilities within environmental states. Our evaluation yields several key insights: UI-TARS stands out as the strongest 7B model, current agents display a bimodal performance pattern in instruction unification, and sub-7B models remain limited in state mastery. We further identify GPT-based planning as a critical bottleneck, and show that grounding data mainly benefits action matching, while trajectory data is more effective for instruction unification.

## 1 Introduction

The recent advancement in vision-language models (VLMs) has spurred increased interest in Device-Control Agents (DC agents), such as utilizing in-the-wild device control to manage graphical user interfaces (GUIs) (Achiam et al., 2023; Anil et al., 2023; Zhang & Zhang, 2023; Hong et al., 2024; Yang et al., 2023). There has been an increasing number of DC agents, making evaluating their capability important.

With the growing complexity and integration of such agents into various applications, effective evaluation methods have become crucial. The current evaluation method for DC agents primarily focuses on the instruction level, providing the current state (e.g., screenshots) and past execution history to determine actions for target instructions (Chen et al., 2024; Deng et al., 2024; Xie et al., 2024; Deng et al., 2023). This method intuitively reveals which instructions DC agents might fail to execute successfully (Rawles et al., 2023; Li et al., 2020; Burns et al., 2022). However, in GUI environments, a single state may contain multiple interactive widgets, each linked to different instructions, presenting an opportunity for diverse actions based on various instruction targets. Evaluating the agent's performance solely at the instruction level may overlook the broader context of these interactions.

To capture a more comprehensive view of agent performance, we propose a new evaluation method based on state-specific instruction accuracy. For each state, we assess the accuracy rate of completing the instructions associated with it, and use the mean accuracy as an indicator of DC agent performance for each state. In a given state, strong agent performance indicates effective learning, whereas poor performance reveals areas where the agent lacks capability and requires improvement. By mapping these strengths and weaknesses across different states, we can precisely delineate the capability boundaries of DC agents. Specifically, our evaluation focuses on the following two research questions:

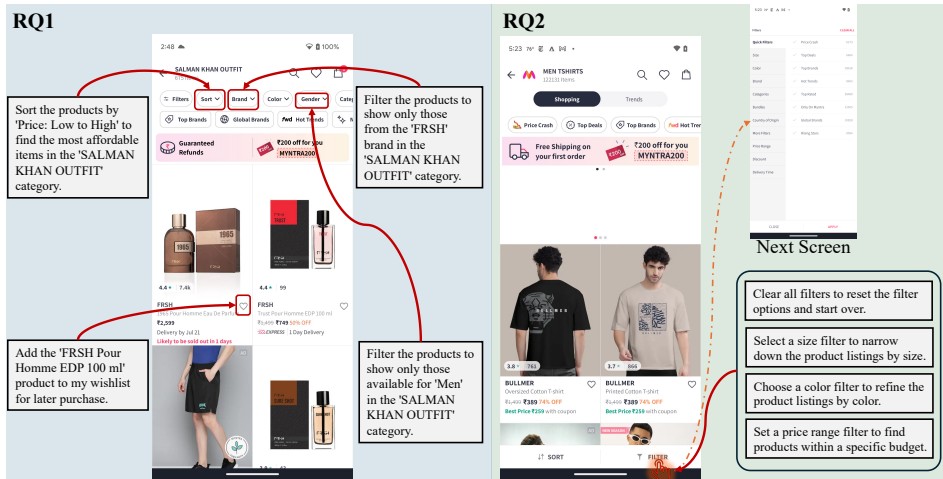

Figure 1: The two examples correspond to the two RQs. The left figure represents RQ1, and the right figure represents RQ2. In RQ1, we explore the interactive widgets in the current state along with their corresponding instructions and actions. In RQ2, we explore the set of instructions that may require the same action.

**RQ1:** Can the agent correctly discriminate and execute multiple distinct instructions mapped to different UI widgets under the same state?

**RQ2:** Can the agent unify semantically diverse instructions into the same action when they should converge on a single widget?

We use the two examples in Figure 1 to illustrate these questions. To address these questions, we introduce an innovative evaluation method, E**X**ploring the Capabilities **BOUND**aries of Device-Control Agent Capabilities (XBOUND). Additionally, we propose two scenarios, Multi-Widget Action Matching and Uni-Widget Instruction Unification, to facilitate state-level analysis. Compared to previous evaluation methods, XBOUND employs a novel Exploration Metric that quantifies the extent to which DC agents master various states. We compute the average action accuracy of instructions associated with each state in these two scenarios. The comparison between XBOUND and existing evaluation methods is shown in Figure 4.

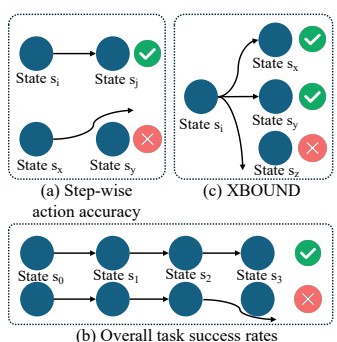

Figure 2: Comparison between XBOUND and existing evaluation methods.

In this work, we evaluate 11 DC agents using the XBOUND method in the Mobile Use domain, aiming to assess their capabilities and limitations across two scenarios systematically. Additionally, we experimentally validate the role of grounding and trajectory data in enhancing agent capabilities across two scenarios. Finally, we summarize four challenging states. This evaluation yields noteworthy insights summarized below:

- Among models below the 7B parameter scale, UI-TARS stands out as the most competitive open-source model, achieving superior performance in both Multi-Widget Action Matching and Uni-Widget Instruction Unification.

- In Uni-Widget Instruction Unification, most current models exhibit a bimodal performance distribution, where DC agents either demonstrate exceptional proficiency or perform poorly.

- Models with fewer than 7B parameters, such as ShowUI and OS-Atlas-4B, demonstrate moderate performance and limited state mastery, indicating that terminal deployment of DC agents remains challenging.

- In Uni-Widget Instruction Unification, UGround performs particularly poorly. Through observations across the three tasks, we find that the overall performance is mainly dragged down by poor planning results from GPT in certain tasks.

- Comparative analysis indicates that grounding data primarily enhances Multi-Widget Action Matching, whereas trajectory data is more effective for improving Uni-Widget Instruction.

- A comprehensive evaluation of DC intelligence requires not only measuring task completion rates but also analyzing fine-grained challenging states such as widget disambiguation, action topology reasoning, and dynamic state understanding.

## 2 RELATED WORK

### 2.1 LLM AS DEVICE-CONTROL AGENTS

Recently, there has been considerable exploration in the field of Device-Control Agents, ranging from box prediction based on HTML and OCR parsing to coordinate prediction based on images (Li & Li, 2022; Li et al., 2024b; Wang et al., 2024a; Zhang et al., 2024b). For example, Yan et al. (2023) utilized the MM-Navigator method to enhance the localization capabilities of GPT-4V. Zheng et al. (2024) proposed a novel prompt method, SeeAct, which combines the reasoning abilities of LLMs with more accurate HTML and OCR parsing to improve GPT-4V's action prediction. Ma et al. (2024) trained a segmented reasoning CoCo-Agent to boost action prediction accuracy. Wu et al. (2024) employed significant engineering effort to collect multi-platform data and train a more powerful Device-Control Grounding Agent OS-Atlas. Qin et al. (2025) trained UI-TARS on large-scale GUI screenshot data, enabling context-aware understanding of UI widgets and precise captioning of interfaces. Gou et al. (2024) introduces a human-like embodiment for DC agents that perceive the environment entirely visually and directly perform pixel-level operations on the GUI.

### 2.2 EVALUATION FOR DEVICE-CONTROL AGENTS

To advance the development of DC Agents, researchers have constructed numerous datasets to evaluate DC agents (Zhou et al., 2023; Xie et al., 2024; Rawles et al., 2024; Lu et al., 2024). Bai et al. (2021); Deka et al. (2017); Cheng et al. (2024) created datasets focused on understanding UI Icons, where models are required to identify the location of relevant UI Icons based on queries. As the development of DC agents progresses, the demands for GUI datasets have shifted, necessitating agents to perform a series of actions in response to user instructions. For example, Rawles et al. (2023); Sun et al. (2022) constructed datasets containing episodes in the form of a sequence of screen-action pairs. Zhang et al. (2024a) supplemented the AITW dataset by adding thought processes. Li et al. (2024a) constructed a fine-grained AndroidControl dataset by including low-level instructions during episodes. Wang et al. (2025) introduced a hierarchical benchmark for evaluating GUI automation agents across six platforms. Zhao et al. (2025) assessed an agent's capability to autonomously generate shortcuts. Zheng et al. (2025) introduced a novel benchmark engineered on the principle of Causal Pathways.

However, current evaluation methods primarily focus on the instruction level and may neglect the broader context of interactions within a given state. To address this limitation, we propose the XBOUND evaluation method, a state-level framework designed to assess agents' capabilities within environmental states.

## 3 NEW METRIC: EXPLORATION METRIC

In this section, we first define the capability of DC agents to clarify how their performance manifests across different scenarios. Alongside this definition, we present the Exploration Metric, outlining its calculation in various scenarios and discussing its significance for evaluating agent capabilities.

### 3.1 CAPABILITY OF DC AGENTS

From the perspective of state evaluation, the capabilities of DC agents are intrinsically linked to action matching and instruction unification. Action matching requires agents to correctly map diverse instructions in the same state to their corresponding UI widgets and execute the intended

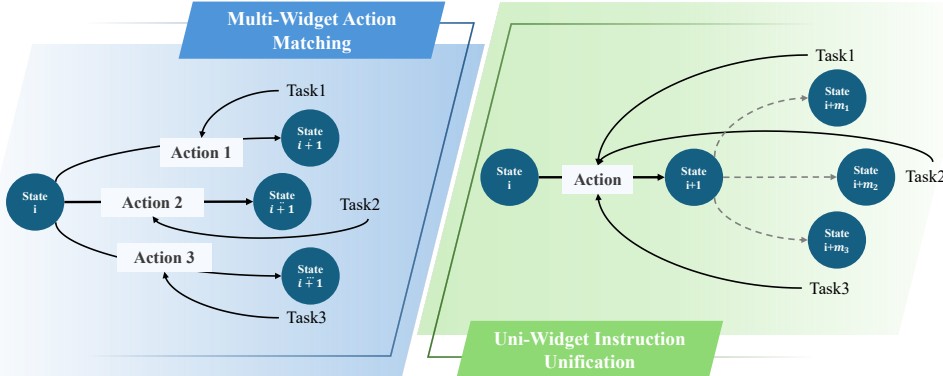

Figure 3: Abstract examples of the two scenarios. Multi-Widget Action Matching: In *State i*, executing the corresponding *Action* under different *Tasks* requirements can transition to different *States i+1*. Uni-Widget Instruction Unification: Executing *Action* in *State i* can transition to *State i+1*, and from *State i+1*, different *Tasks* can lead to different *States i+m*.

actions, reflecting their ability to discriminate between multiple instruction–action pairs. In contrast, instruction unification requires agents to interpret semantically diverse instructions that should converge to the same UI action, reflecting their ability to comprehend instruction-level semantics and generalize across varied command expressions. Consequently, we categorize the capability of DC agents into two scenarios: Multi-Widget Action Matching and Uni-Widget Instruction Unification.

**Multi-Widget Action Matching:** In this scenario, a set of instructions is collected under the same state, where each instruction corresponds to a distinct UI widget and requires executing its associated action. This setting evaluates whether the agent can accurately perceive the relevant UI widgets and correctly match each instruction to the intended action, thus reflecting its capability of instruction–action discrimination.

**Uni-Widget Instruction Unification:** In this scenario, a set of diverse instructions is provided under the same state, but all instructions should converge to the same target action on a single widget. This setting evaluates whether the agent can understand the semantic equivalence of varied instruction expressions and unify them into a consistent action, thus reflecting its capability of instruction-level semantic comprehension.

Figure 3 presents examples of these two scenarios. By dividing these two scenarios, we calculate the agents' performance in different states to identify their capability boundaries.

## 3.2 EXPLORATION METRIC

To evaluate the capabilities of DC agents within these two scenarios, we introduce the XBOUND evaluation method, which measures agent performance along two dimensions: MWAM and UWIU. The **MWAM** dimension aligns with **Multi-Widget Action Matching**, assessing whether agents can correctly map instructions to their corresponding UI widgets and generalize these behaviors across diverse visual contexts. The **UWIU** dimension corresponds to **Uni-Widget Instruction Unification**, evaluating whether agents can consistently execute the same action when faced with semantically varied instructions, thereby reflecting their robustness in handling diverse task formulations.

XBOUND introduces a novel Exploration Metric for quantifying agent capability within environmental states. We gather the set of executable instructions within the same state and calculate the average accuracy of this instruction set as a measure of the agent's capability in the given state. By evaluating performance across all states, we derive corresponding values for each state's agent performance, ultimately illustrating the agent's capability boundaries. The formulas for the Exploration Metric(EM) are as follows:

$$EM_{state} = \frac{1}{m} \sum_{i=1}^{m} I(A_i), \tag{1}$$

$$EM_{all} = \frac{1}{s} \sum_{j=1}^{s} EM_{state_j}, \qquad (2)$$

where $I(\cdot)$ is the indicator function, which equals 1 if the action $A_i$ is correct and 0 otherwise. The variable m represents the number of instructions associated with each screenshot, while s denotes the number of screenshots.

Since current benchmarks cannot directly utilize the XBOUND evaluation technique, it is necessary to expand the existing dataset to meet our requirements. Owing to community development in recent years, current benchmarks provide detailed accessibility trees information, which we leverage to improve the Android Control dataset (Li et al., 2024a). The detailed pipeline is provided in the Appendix A.3. Finally, we retain 43,759 instructions in the MWAM dimension, while the UWIU dimension contains 13,460 instructions.

## 4 EXPERIMENT

The experimental setup is described in Sec. 4.1; overall evaluation results are presented in Sec. 4.2; task-specific performance is detailed in Sec. 4.3; error analysis is provided in Sec. 4.4; the comparison between grounding training and trajectory training is discussed in Sec. 4.5; and the summary of challenging states is included in Sec. 4.6.

### 4.1 EXPERIMENTAL SETUP

**DC agent.** We select eleven open-source DC agents with fewer than 7B parameters as evaluation models, including ShowUI (Lin et al., 2024), SeeClick (Cheng et al., 2024), Qwen2-VL-Instruct (Wang et al., 2024b), Uground (Gou et al., 2024), OS-Atlas (Wu et al., 2024), Aguvis (Xu et al., 2024), GUI-Owl (Ye et al., 2025), and UI-TARS (Qin et al., 2025). We adhere to the prompts they utilize while deliberately excluding execution history from the inputs. Our experiments are conducted on an A100 GPU with 80GB of memory.

**Evaluation Metrics.** In line with the criteria set forth by Zhang & Zhang (2023); Wu et al. (2024), an action is considered correct if its type matches the ground-truth type. Specifically, for CLICK and LONG PRESS actions, correctness in the UWIU dimension is determined if they occur within a 14% screen distance from the reference gestures. In the MWAM dimension, correctness is assessed based on whether the actions fall within the bounding box of the ground truth UI icon. For SCROLL actions, correctness is evaluated by checking if the direction (up, down, left, or right) matches the reference gesture. For TYPE actions, correctness is assessed using the F1 score; the action is deemed correct if the score is below a threshold of 0.5, as set in our experiments.

### 4.2 COMPREHENSIVE EVALUATION

We calculate the results of the Exploration Metric corresponding to each state. To enhance state analysis, we partition the Exploration Metric into four distinct intervals:

- **Learning Stage** ($EM_{state} < 30\%$)
  The current state suggests that DC agents are still in the learning and adaptation phase, indicating unfamiliarity with the environment.

- **Improvement Stage** ($30\% \leq EM_{state} < 60\%$)
  The current state indicates that DC agents have started to grasp certain operations and are making progress.

- **Proficient Stage** ($60\% \leq EM_{state} < 90\%$)
  The current state signifies that DC agents possess a relatively proficient understanding and can perform most actions.

- **Expert Stage** ($90\% \leq EM_{state} \leq 100\%$)
  The current state implies that DC agents have achieved a comprehensive and expert level of understanding.

Table 1: The assessment of DC agents' capabilities spans two dimensions: MWAM and UWIU. Specifically, the percentage of states is reported for the four stages, i.e., Learning Stage (LS), Improvement Stage (IS), Proficient Stage (PS), and Expert Stage (ES). The Exploration Metric (EM) quantifies the overall mastery of states by DC agents, reflecting their accuracy in completing instructions within each state. In contrast, the Success Rate (SR) measures the step-wise success rate of DC agents across the dataset, indicating their proficiency in executing individual steps. For each DC agent, the best-performing Stage results are highlighted in bold, and the second-best are underlined. For the EM and SR, we bold the best-performing results of the 11 DC agents.

| Model | MWAM | | | | | | UWIU | | | | | |
|---|---|---|---|---|---|---|---|---|---|---|---|---|
| | LS | IS | PS | ES | EM | SR | LS | IS | PS | ES | EM | SR |
| ShowUI-2B | **75.09** | 19.17 | 4.17 | 1.57 | 18.51 | 19.70 | **68.44** | 9.76 | 5.77 | 16.03 | 25.27 | 25.54 |
| OS-Atlas-4B-Pro | **37.29** | 34.25 | 22.51 | 5.95 | 41.92 | 42.82 | **57.16** | 19.60 | 7.25 | 15.99 | 31.80 | 30.24 |
| OS-Atlas-7B-Pro | 20.41 | 27.74 | **35.48** | 16.37 | 57.59 | 59.50 | 36.83 | 12.95 | 12.90 | **37.32** | 53.44 | 53.22 |
| SeeClick | **53.16** | 24.68 | 15.97 | 6.19 | 32.58 | 33.72 | **77.05** | 10.60 | 3.42 | 8.93 | 17.15 | 14.93 |
| Qwen2-VL-7B-Ins | 26.05 | **35.00** | 30.98 | 7.97 | 49.30 | 50.88 | **58.84** | 14.85 | 10.21 | 16.11 | 31.52 | 32.19 |
| Aguvis-7B | 19.23 | 29.16 | **37.25** | 14.35 | 57.39 | 59.23 | **37.32** | 15.91 | 12.95 | 33.91 | 51.29 | 51.47 |
| UGround-7B | 14.20 | 25.53 | **38.11** | 22.16 | 63.00 | 63.66 | **75.95** | 3.39 | 1.90 | 18.81 | 21.97 | 20.35 |
| UI-TARS-7B-SFT | 10.54 | 24.40 | **42.52** | 22.55 | 66.96 | 68.20 | **39.41** | 13.98 | 11.97 | 34.64 | 50.53 | 52.40 |
| UI-TARS-7B-DPO | 11.45 | 24.59 | **42.20** | 21.76 | 66.08 | 67.57 | **37.55** | 14.03 | 13.13 | 35.29 | 52.02 | 53.41 |
| UI-TARS-1.5-7B | 13.54 | 25.33 | **40.24** | 20.89 | 64.25 | 65.82 | 16.66 | 8.25 | 10.03 | **65.05** | 76.44 | 79.69 |
| GUI-Owl-7B | 14.13 | 25.97 | **41.24** | 18.65 | 62.97 | 64.87 | 32.03 | 14.33 | 14.59 | **39.05** | 56.96 | 58.41 |

DC agents with stronger capabilities should have a higher proportion in the Proficient Stage and Expert Stage, and a lower proportion in the Learning Stage and Improvement Stage. We present the four stages and the Exploration Metric of various DC agents across different dimensions in Table 1.

**Performance of models with fewer than 7 billion parameters reveals insufficient proficiency.** Analysis of ShowUI and OS-Atlas-4B-Pro indicates that agents with parameter scales smaller than 7 billion still exhibit inadequate performance. Across both MWAM and UWIU dimensions, most state performances remain within the Learning Stage and Improvement Stage. Specifically, ShowUI-2B is in the Learning Stage as high as 75.09%, highlighting the challenges of achieving effective terminal deployment with smaller models.

**UI-TARS models demonstrate superior performance among the 7 billion parameter models.** Observations from the table show that the UI-TARS series is the top performer within this parameter range. Each of the three UI-TARS models achieves at least 64% overall Exploration Metric performance in the MWAM dimension, with most state performances in the Proficient Stage and Expert Stage. In the UWIU dimension, UI-TARS-1.5-7B records the best performance. Given the unknown specifics of their training data, we speculate that version 1.5 includes more trajectory tasks, enhancing the model's comprehension of current action execution concerning future tasks.

**UGround exhibits anomalous performance in the UWIU dimension.** Analysis indicates that UGround's errors largely stem from incorrect planning by GPT, primarily due to erroneous environmental perception. For example, the model may plan to return to the desktop to open an email app when attempting to forward content within an app. This highlights the necessity of training the model with relevant planning data.

**A bimodal distribution phenomenon observed in the UWIU dimension.** In the UWIU dimension, a bimodal distribution emerges where agents exhibit complete absence or presence of action learning. This suggests that current DC agents are yet to achieve human-like intelligence and still have significant developmental strides to make.

## 4.3 CAPABILITY EVALUATION BASED ON TASK

We select three tasks from the test data and sample 2,000 instructions for each task. Correct instructions are manually filtered, and highly repetitive instructions are removed. The final statistical

Table 2: The proportion of states in the four stages for 6 agents across three tasks. The best-performing Stage results are highlighted in bold, and the second-best are underlined.

| Task | Model | MWAM | | | | UWIU | | | |
|------|-------|------|------|------|------|------|------|------|------|
| | | LS | IS | PS | ES | LS | IS | PS | ES |
| Maps | Aguvis-7B | 6.90 | 24.14 | **37.93** | 31.03 | **40.92** | 13.64 | 18.18 | 27.27 |
| | UGround-7B | 0 | 27.59 | **51.72** | 20.69 | 43.75 | 0 | 0 | **56.25** |
| | UI-TARS-7B-SFT | 0 | 17.24 | 37.93 | **44.83** | 40.91 | 0 | 0 | **59.09** |
| | UI-TARS-7B-DPO | 3.45 | 10.34 | 37.93 | **48.28** | 45.45 | 0 | 0 | **54.55** |
| | UI-TARS-1.5-7B | 3.45 | 10.34 | 34.48 | **51.72** | 9.09 | 9.09 | 9.09 | **72.73** |
| | GUI-Owl-7B | 3.45 | 6.90 | 27.59 | **62.07** | 27.27 | 13.64 | 22.73 | **36.36** |
| News | Aguvis-7B | 4 | 24 | 32 | **40** | 8 | 28 | 4 | **60** |
| | UGround-7B | 4 | **36** | 28 | 32 | **73.68** | 0 | 0 | 26.32 |
| | UI-TARS-7B-SFT | 0 | 16 | 32 | **52** | **56** | 0 | 0 | 44 |
| | UI-TARS-7B-DPO | 4 | 12 | 32 | **52** | **56** | 0 | 0 | 44 |
| | UI-TARS-1.5-7B | 4 | 12 | **44** | 40 | 0 | 4 | 8 | **88** |
| | GUI-Owl-7B | 0 | 20 | 24 | **56** | 12 | 24 | 8 | **56** |
| Shopping | Aguvis-7B | 4.65 | 11.63 | **53.49** | 30.23 | 21.88 | 28.12 | 3.12 | **46.88** |
| | UGround-7B | 11.63 | 23.26 | **34.88** | 30.23 | **84.62** | 0 | 0 | 15.38 |
| | UI-TARS-7B-SFT | 2.33 | 9.3 | 37.21 | **51.16** | 43.75 | 0 | 0 | **56.25** |
| | UI-TARS-7B-DPO | 2.33 | 6.98 | 39.53 | **51.16** | 43.75 | 0 | 0 | **56.25** |
| | UI-TARS-1.5-7B | 2.33 | 6.98 | 34.88 | **55.81** | 9.38 | 9.38 | 0 | **81.25** |
| | GUI-Owl-7B | 2.33 | 9.3 | 30.23 | **58.14** | 25 | 6.25 | 12.5 | **56.25** |

results are presented in the Appendix A.3. We evaluate the performance of 11 agents on these three tasks. The proportion of states in each of the four stages for six agents is shown in Table 2; results for the remaining five agents are presented in Appendix A.4. The EM capability chart is provided in Figure 4. Analysis of Table 2 reveals the following:

The UI-TARS and GUI-Owl series currently demonstrate the best performance in the MWAM dimension (Multi-Widget Action Matching) for the Maps, News, and Shopping tasks.

UGround performs well on the Maps task but poorly on News and Shopping. This indicates that employing GPT as a planning model does not consistently result in poor performance across all tasks; rather, certain tasks adversely affect its overall performance.

UGround, UI-TARS-7B-SFT, and UI-TARS-7B-DPO exhibit a pronounced bimodal distribution in the UWIU dimension for Maps, News, and Shopping tasks, suggesting that these models fail to train on certain actions and possess learning blind spots.

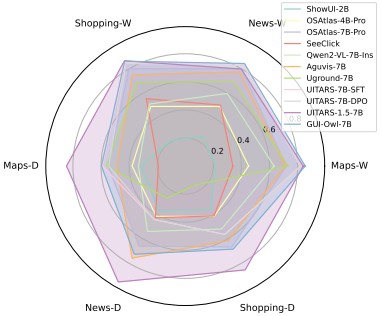

Figure 4: The EM of 11 agents across three tasks.

Figure 4 shows that UI-TARS-1.5-7B is the best-performing agent overall across the three tasks, while Aguvis, UI-TARS-7B-SFT, UI-TARS-7B-DPO, and GUI-Owl display comparable performance.

## 4.4 ERROR ANALYSIS

By calculating the Exploration Metric, we can quickly identify states with low accuracy. In further analyzing these poor performance states, we look closely at the actions undertaken by the agent and compute their similarity. Typically, the more similar the actions, the closer the similarity value approaches 1; conversely, the more disparate the actions, the closer the similarity value approaches 0.

Table 3: The comparison between grounding training and trajectory training. The best-performing Stage results are highlighted in bold, and the second-best are underlined. For the EM, we bold the best-performing results.

| Model | MWAM | | | | | UWIU | | | | |
|---|---|---|---|---|---|---|---|---|---|---|
| | LS | IS | PS | ES | EM | LS | IS | PS | ES | EM |
| Qwen2-VL-7B-Ins | 26.05 | **35.00** | 30.98 | 7.97 | 49.30 | **58.84** | 14.85 | 10.21 | 16.11 | 31.52 |
| OS-Atlas-7B-Base | 16.07 | 27.40 | **39.26** | 17.27 | **60.61** | **52.14** | 17.93 | 9.22 | 20.71 | 36.91 |
| OS-Atlas-7B-Pro | 20.41 | 27.74 | **35.48** | 16.37 | 57.59 | 36.83 | 12.95 | 12.90 | **37.32** | **53.44** |

If the similarity value of erroneous actions nears 1, this indicates systematic errors within these states. We explain the cause of mistakes through the two scenarios (Section 3.1). (1) Multi-Widget Action Matching: In this scenario, the agent may execute incorrect actions (such as press_back and press_home) due to insufficient understanding of the state, implying that the current state is unfamiliar to the agent. (2) Uni-Widget Instruction Unification: This may be due to the agent having learned only the action associated with the current state, without recognizing distinctions between task instructions, which leads it to execute the same action regardless of the task. This indicates that the model possesses limited capability to handle diverse instructions in that state.

If the similarity value of erroneous actions is closer to 0, it signifies inadequate learning by the agent in these states. We explain the cause of mistakes through the two scenarios. (1) Multi-Widget Action Matching: In this scenario, the current state usually contains UI widgets unlearned by the agent, leading to incorrect actions, evidencing insufficient action matching capability for the current state. (2) Uni-Widget Instruction Unification: The agent fails to grasp the relationship between the current action and future state transitions, resulting in errors in action selection. This indicates inadequate learning of actions by the agent. More examples are presented in the Appendix A.5.

### 4.5 GROUNDING TRAINING AND TRAJECTORY TRAINING COMPARISON

In this section, we discuss the impact of grounding training and trajectory training on Mobile Use Agents. Utilizing the XBOUND evaluation method, we assess three models: Qwen2-7B-VL-Instruct, OS-Atlas-7B-Base, and OS-Atlas-7B-Pro. OS-Atlas-7B-Base is trained with grounding data based on Qwen2-7B-VL-Instruct, whereas OS-Atlas-7B-Pro incorporates trajectory data based on OS-Atlas-7B-Base. The results are presented in Table 3.

Comparing the Base model with the Qwen model reveals that grounding data enhanced the Base model's performance in the MWAM dimension, improving its understanding of Multi-Widget Action Matching. However, significant improvement in the UWIU dimension is not observed until trajectory data is employed for training, which subsequently enhances performance within Uni-Widget Instruction Unification. This also demonstrates that grounding data is associated with the agents' action-matching abilities, whereas trajectory data is linked to their decision-making abilities.

### 4.6 CHALLENGING STATES

During the XBOUND evaluation process, we identify several challenging states that require a certain level of intelligence from DC agents. We categorize these states into the following four demands:

(1) **Understanding of UI Icons:** Due to limitations in current grounding and trajectory data, many UI icons are absent from agents' training, potentially causing agents to fail in learning tasks associated with certain UI icons. This scenario heavily assesses the agents' action-matching abilities regarding UI icons. In Figure 5(a), the agent fails to recognize the image-search icon and repeatedly selects the text search box instead.

(2) **Distinction Between Similar UI Icons:** Tasks in domains such as shopping and media often present multiple visually similar icons on the same page. Agents must correctly align instructions with the intended target, testing their widget-level discrimination ability. In Figure 5(b), the agent misinterprets a "like" command and clicks on the wrong post.

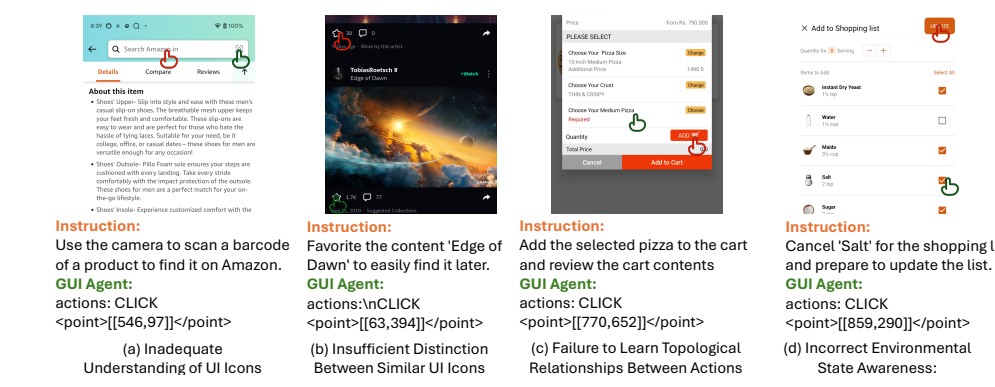

Figure 5: Four types of challenging states are illustrated through representative examples. A green pointer indicates the correct action, while a red pointer denotes the incorrect action.

(3) **Topological Relationships Between Actions:** In specific task scenarios on certain pages, actions may have topological dependencies where action B must precede action A. This scenario requires agents to comprehend the topological order of actions and possess a deeper understanding of their meanings. In Figure 5(c), the instruction is to complete the customization of a pizza directly. However, the "Choose Your Medium Pizza" option isn't selected, and "Add" should only be executed after completing the "Choose Your Medium Pizza" step.

(4) **Environmental State Awareness:** For some ordering and shopping tasks, product requirements often need alteration. This scenario demands that agents understand or perceive the environmental state necessary for task execution correctly. In Figure 5(d), the goal is to remove "Salt" before the update, but the DC agent fails to notice that "Salt" is already added and proceeds with the update action instead.

## 5 LIMITATIONS AND FUTURE WORK

This work presents a novel evaluation method, which we extend with the Android Control dataset since it does not directly apply to existing data. While LLM assists in filtering, the resulting data may not fully ensure the accuracy of instructions and actions. Despite this, **the XBOUND method highlights promising directions for future research.**

**More Complete and Refined Evaluation:** The evaluation data in this work have the following characteristics: offline data and trajectory independence, which make it impossible to build a full trajectory tree. In future development of evaluation data for Mobile Use, we can organize app-level trajectory tree data and evaluate DC agents' capabilities app-wise using the XBOUND method.

**New Ideas for Enhancing DC Agents' Capabilities:** Currently, improving DC agents' performance primarily focuses on the instruction level, relying on extensive trajectory data collection to enhance agents' capabilities. In the future, if augmented instructions cease to be efficient in enhancing agents' capabilities, we can evaluate agents' performance across different states and focus on improving performance in underperforming states to boost their overall capabilities.

## 6 CONCLUSION

This study delineates the scenarios of DC agents' capability within states and introduces a novel evaluation method, XBOUND. XBOUND provides a state-level evaluation framework, serving as a tool to assess agents' capabilities within environmental states. From the perspective of state evaluation, we define two distinct scenarios, evaluate the performance of 11 open-source agents within the Mobile Use domain, offering new insights at the state level. Our evaluation reveals several key insights. UI-TARS emerges as the most effective model at the 7B scale. Current agents exhibit a bimodal performance pattern in instruction unification, while sub-7B models demonstrate limited state mastery. In addition, we identify GPT-based planning as a critical bottleneck and show that grounding data primarily improves action matching, whereas trajectory data proves more effective for instruction unification.

## 7 ETHICS STATEMENT

We confirm that this work adheres to the ethical guidelines set forth by the ICLR 2026 conference.

## 8 REPRODUCIBILITY STATEMENT

We hereby declare that our work is fully reproducible. Using our data augmentation pipeline, multiple instructions can be generated. By applying these instructions (or those we provide) to evaluate the 11 DC agents, the same observations can be obtained in both the Multi-Widget Action Matching and Uni-Widget Instruction Unification scenarios.

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

# A APPENDIX

## A.1 THE USE OF LARGE LANGUAGE MODELS

In this work, LLMs serve two primary purposes: one is for polishing the paper, and the other is for data augmentation.

## A.2 TRAJECTORY TREE DATASET

Common GUI trajectory datasets record a series of action sequences that result in screen transitions. However, trajectory trees focus on other possible actions and tasks that may branch out from a given state. Formally, given the state $S_t$ at the time step $t$ and various user instructions $I_t^1, I_t^2, \ldots I_t^M$, the DC agent $G$ will take corresponding actions under different instructions, such as $A_t^1 = G(S_t, I_t^1)$, $A_t^2 = G(S_t, I_t^2)$, ..., $A_t^M = G(S_t, I_t^M)$, where $M$ represents the number of instructions. Each instruction completes a corresponding trajectory sequence $E = \{(S_t, A_t)_{t=1}^T, I\}$, where $T$ represents the total steps. These trajectory sequences are constructed into a trajectory tree dataset $T = \{(S_t^m, (I^m, A^m), S_{t+1}^m)_{t=1}^T\}_{m=1}^M$ based on overlapping states.

## A.3 DATA EXPANSION METHOD

Using the accessibility trees provided, we commence by annotating each screenshot with accessibility trees from Android Control to identify clickable and visible UI icons, which are sequentially numbered. Red boxes highlight these icons on screenshots, and their numbers ensure clear identification. Subsequently, GPT-4omini is employed to generate both high-level and low-level instructions. After acquiring the instructions, we utilize Qwen2.5-VL-72B-Instruct to translate them into corresponding actions (e.g., click, scroll, type, etc.).

In the MWAM dimension, we focus on collecting single screenshots and request GPT-4omini to produce task instructions based on the UI icons present on these screenshots. In the UWIU dimension, we primarily gather screenshot-action-screenshot pairs $(S_t, A_t, S_{t+1})$ and ask GPT-4omini to expand instructions for the UI icons on the subsequent screenshot $S_{t+1}$. We then combine the high-level instructions $I_{t+1}^1, I_{t+1}^2, \ldots, I_{t+1}^M$ collected from screenshot $S_{t+1}$ with action $A_t$ and screenshot $S_t$ into the structure $\{(S_t, (I^m, A_t), S_{t+1})\}_{m=1}^M$. When DC agents strive to execute the expanded instructions from the subsequent screenshot, they must perform the collected actions on the previous screenshot.

To ensure data quality, GPT-4omini evaluates whether actions and low-level instructions satisfy the high-level instructions, resulting in a dataset of successful interactions. Since high-level instructions are generated per screenshot and do not constitute a complete trajectory, the test dataset is termed a "pseudo" trajectory tree dataset. Figure 6 visually represents the dataset construction process for these dimensions.

Our main experiment dataset comprises 1,536 episodes with 43,759 instructions, where the MWAM dimension includes 43,759 instructions, and the UWIU dimension contains 13,460 instructions. We have tallied the number of instructions associated with each screenshot and the action distribution, with detailed information presented in Figure 7 and Figure 8.

Following Li et al. (2024a), we use Qwen2-vl-7B-Instruct to classify the test data based on the app categories provided. Ultimately, we select the three most prevalent app categories within the test set for further analysis. The statistical results of the collected instructions are presented in Table 4.

Table 4: Data statistics of the three tasks in both dimensions.

| Task | Width | UWIU |
|---|---|---|
| Maps | 135 | 73 |
| News | 115 | 58 |
| Shopping | 205 | 99 |

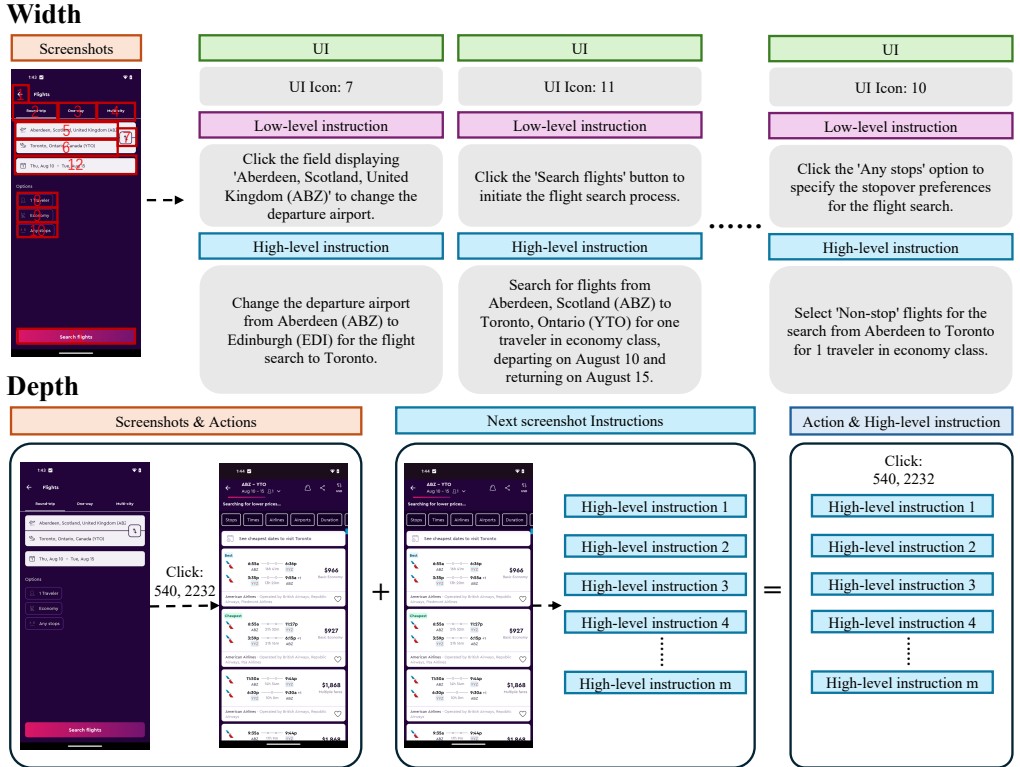

Figure 6: The data collection construction process involves MWAM and UWIU dimensions. **MWAM Dimension:** Screenshots are annotated, and GPT4o-mini is utilized to select UI elements for generating both low-level and high-level instructions. **UWIU Dimension:** High-level instructions corresponding to subsequent screenshots are identified based on transitions between screenshots, alongside the collection of actions and high-level instructions.

## A.4 ANOTHER 5 AGENT CAPABILITY PERFORMANCE

The performance of the remaining five agents on three different tasks is reported in Table 5.

We observe that the 2B model still performs poorly, with ShowUI remaining the weakest across all three tasks. In contrast, OS-Atlas-4B-Pro has already outperformed SeeClick and is approaching the performance of Qwen2-VL-7B-Ins.

## A.5 SIMILARITY CASE

Figure 9 illustrates an example from the Multi-Widget Action Matching scenario where the similarity of incorrect actions is 0, while Figure 10 presents a case from the same scenario where the similarity of incorrect actions is 1. Figure 11 illustrates an example from the Uni-Widget Instruction Unification scenario where the similarity of incorrect actions is 0, while Figure 12 presents a case from the same scenario where the similarity of incorrect actions is 1.

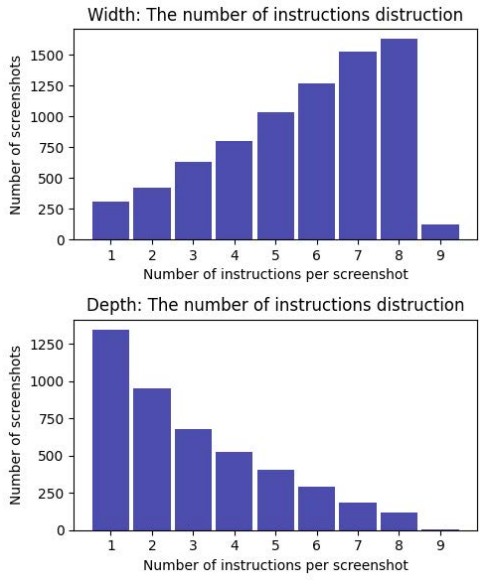

Figure 7: Instructions per screenshot distribution

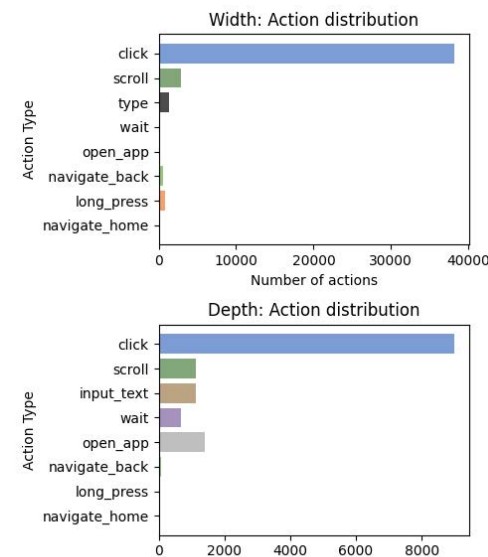

Figure 8: Action distribution

Table 5: The proportion of states in the four stages for 11 agents across three tasks.

| Task | Model | Width | | | | UWIU | | | |
|---|---|---|---|---|---|---|---|---|---|
| | | LS | IS | PS | ES | LS | IS | PS | ES |
| Maps | ShowUI-2B | **68** | 24 | 8 | 0 | **70** | 0 | 0 | 30 |
| | OS-Atlas-4B-Pro | 31.03 | 27.59 | **37.93** | 3.45 | **54.55** | 18.18 | 9.09 | 18.18 |
| | OS-Atlas-7B-Pro | 17.24 | 10.34 | 24.14 | **48.28** | 36.36 | 13.64 | 13.64 | 34.36 |
| | SeeClick | **44.83** | 34.48 | 17.24 | 3.45 | **81.25** | 0 | 6.25 | 12.5 |
| | Qwen2-VL-7B-Ins | 4 | **44** | 36 | 16 | **57.14** | 14.29 | 9.52 | 19.05 |
| News | ShowUI-2B | **64** | 24 | 8 | 4 | **48** | 24 | 4 | 24 |
| | OS-Atlas-4B-Pro | 20 | 20 | **36** | 24 | **48** | 28 | 0 | 24 |
| | OS-Atlas-7B-Pro | 4 | 16 | 32 | **48** | 20 | 20 | 12 | **48** |
| | SeeClick | 24 | **36** | 28 | 12 | **52.63** | 10.53 | 0 | 36.84 |
| | Qwen2-VL-7B-Ins | 12 | 28 | **48** | 12 | 36 | 20 | 4 | **40** |
| Shopping | ShowUI-2B | **67.44** | 30.23 | 2.33 | 0 | **59.38** | 12.5 | 0 | 28.12 |
| | OS-Atlas-4B-Pro | 25.58 | 25.58 | **44.19** | 4.65 | **46.88** | 15.62 | 12.5 | 25 |
| | OS-Atlas-7B-Pro | 2.33 | 13.95 | **41.86** | 41.86 | 21.88 | 15.62 | 12.5 | **50** |
| | SeeClick | 28.6 | 30.23 | **37.21** | 13.95 | **50** | 19.23 | 3.85 | 26.92 |
| | Qwen2-VL-7B-Ins | 23.26 | 32.56 | **39.53** | 4.65 | 34.38 | 25 | 3.12 | **37.5** |

A.6 PROMPT

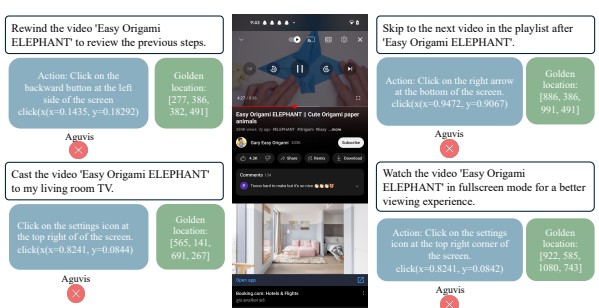

Figure 9: Similarity=0 in the MWAM dimension.

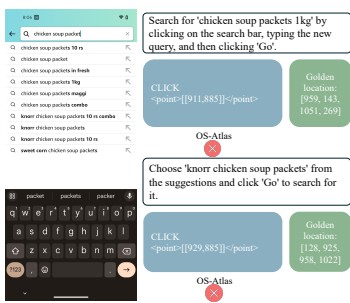

Figure 10: Similarity=1 in the MWAM dimension.

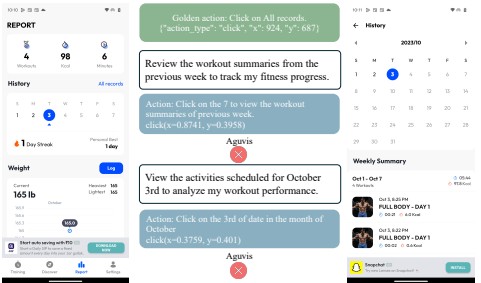

Figure 11: Similarity=0 in the UWIU dimension.

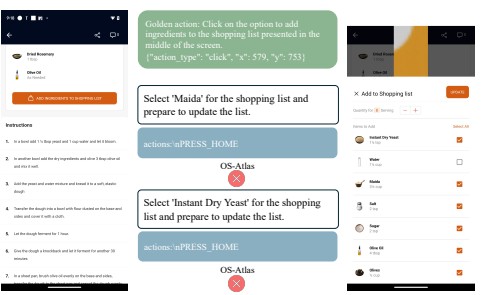

Figure 12: Similarity=1 in the UWIU dimension.

864
865
866
867
868
869
870
871
872
873
874
875
876
877
878
879
880
881
882
883
884
885
886
887
888
889
890
891
892
893
894
895
896
897
898
899
900
901
902
903
904
905
906
907
908
909
910
911
912
913
914
915
916
917

**Prompt for constructing the trajectory tree dataset.**

You are a mobile expert who excels at interacting with elements on mobile screens to complete tasks. I have a task for you, and I hope you can use your extensive knowledge to identify interactive elements on mobile screens. I will provide you with the following information:

1. The type of action currently being executed, which can be one of five types: CLICK, SCROLL, TYPE, PRESS_BACK, and LONG_PRESS. You need to choose an action that can interact with the current screen.

2. Analysis of the mobile screen, which corresponds to the marked boxes in the images. Your task is to identify five interactive elements on the current mobile screen. The output should include four parts:

1. Sub-Instruction: Identify the interactive elements and generate natural language instructions for interacting with these elements. The instructions should be concise, clear, and executable, and must include critical details such as filenames, times, or other content as they appear on the screen. For example: "Scroll left to open the app drawer, displaying all installed applications on the device", "Click the chat interface, allowing the user to view and participate in conversation", "Type the username 'Agent', preparing for the next step in logging into the account".

2. Analysis: Analyze possible subsequent operations based on the current interface and action instructions. This analysis should involve step-by-step reasoning, considering potential changes on the screen and actions that can be taken after these changes. For example: "After clicking the plus button, a dropdown menu appears with an option to create a document. I can select this option to create a new document. First, I need to name the document, then enter content into the document, and finally save the document and exit".

3. High-Level Instruction: Based on the analysis results, envision a high-level task that can be completed within the current interface. There are two types of High-Level Instructions: Task-Oriented: Completing a series of operations to achieve a specific goal. Question-Oriented: Performing a series of operations and deriving an answer to a specific question. For example: Share my favorite Book "the Queen's Gambit" to my Friend Natalie larson over her gmail address -natalie.larson1998@gmail.com from the PocketBook app. Ensure that the High-Level Instruction is executable by including all critical specifics, such as filenames, relevant timings, or required details.

4. UI item: Based on the current page parsed result and action instructions, identify the corresponding UI item and provide the specific number.

You only need to return a dictionary formatted as follows: { "Sub-Instruction": "xxx", "Analysis": "xxx", "High-Level-Instruction": "xxx", "UI item": x }

Current screen analysis:

Figure 13: Prompt for constructing the trajectory tree dataset.

Prompt for reasoning the correct golden action.

You are a GUI task expert, I will provide you with a low-level instruction, a golden ui with its corresponding ID.
Low-level instruction:
UI ID:
Please generate the action for the next step.
Candidate Actions:
"action_type": "click", "ui": ¡ui_idx¿
"action_type": "long_press", "ui": ¡ui_idx¿
"action_type": "type", "text": ¡text_input¿
"action_type": "scroll", "direction": ¡up, down, left, or right¿
"action_type": "navigate_home"
"action_type": "navigate_back"
"action_type": "open_app", "app_name": ¡app_name¿
"action_type": "wait"
"action_type": "status", "goal_status": ¡"successful","infeasible"¿
You need to generate a script in the form: actions: ACTION
Make sure to consider the details in the screenshot and the task requirements to create an accurate and functional script."

Figure 14: Prompt for evaluating whether actions correctly execute low-level instructions.

Prompt for reasoning the correct golden action.

You are a mobile expert who excels at interacting with elements on mobile screens to complete tasks. I have a task for you, and I hope you can use your extensive knowledge to identify interactive elements on mobile screens. I will provide you with the following information:
1. A low-level instruction, which we will follow to perform actions on the current screen.
2. The type of action currently being executed, which can be one of two types: CLICK or LONG_PRESS. You need to determine whether this action can fulfill the current low-level instruction.
3. The current screen environment, with the position where the action(click and long_press) needs to be executed marked by a red box.
I will provide you with a screenshot, along with the low-level instructions and the action to be executed. Your task is to determine whether the current action brings us closer to achieving the low-level instruction. If the current action contributes to the realization of the low-level instruction, answer "Yes"; otherwise, answer "No".
You only need to return a dictionary formatted as follows: { "Analysis": "xxx", "Correct": Yes/No }

Figure 15: Prompt for evaluating whether actions correctly execute low-level instructions.

**Prompt for evaluating whether low-level instructions solve high-level instructions.**

You are a mobile expert who excels at interacting with elements on mobile screens to complete tasks. I have a task for you, and I hope you can use your extensive knowledge to identify interactive elements on mobile screens. I will provide you with the following information:

1. A high-level instruction, which is our ultimate goal to be executed.

2. A low-level instruction, which we will follow to perform actions on the current screen.

3. The current screen environment, with the position where the action needs to be executed marked by a red dot.

I will provide you with a screenshot, along with the high-level and low-level instructions to be executed. Your task is to determine whether the current low-level instruction brings us closer to achieving the high-level instruction. If the current low-level instruction contributes to the realization of the high-level instruction, answer "Yes"; otherwise, answer "No".

You only need to return a dictionary formatted as follows: { "Analysis": "xxx", "Correct": Yes/No }

Figure 16: Prompt for evaluating whether low-level instructions solve high-level instructions.

> **Prompt for classifying trajectories into specific app tasks.**
>
> You are a GUI agent. I'll give you a total goal, a screenshot, and some categories of apps, and I'll ask you to choose the closest to the general goal among those categories.
> ## Output Format
> You only need to return a dictionary formatted as follows: { "Analysis": "xxx", "Categories": "xxx" }
> ## APP Categories
> 1. Shopping
> 2. Productivity & Office
> 3. Other
> 4. Files
> 5. Transportation
> 6. Health & Fitness
> 7. Recipes
> 8. Flights
> 9. Clock & Alarms
> 10. Reminders
> 11. Voice recording
> 12. Education
> 13. Books
> 14. Email
> 15. Calendar
> 16. Notes & Todos
> 17. Maps
> 18. Videos
> 19. News
> 20. Meditation
> 21. Weather
> 22. Finance
> 23. Art & crafts
> 24. Gardening
> 25. Contacts
> 26. Drawing
> 27. Music
> 28. Real estate
> 29. Messaging
> ## Total Goal

Figure 17: Prompt for classifying trajectories into specific app tasks.

