# OpenReview forum: "XBOUND: Exploring Capability Boundaries of Device-Control Agents at the State Level"
_ICLR.cc/2026/Conference — ICLR 2026 Conference Withdrawn Submission_

### Official Review · Reviewer_bq9A · 2025-10-15

**Soundness:** 2
**Presentation:** 3
**Contribution:** 2
**Rating:** 2
**Confidence:** 5

**Summary:**

This paper presents XBOUND (Exploring Capability Boundaries), a novel evaluation framework for device-control agents that operate graphical user interfaces. Instead of measuring performance at the instruction level, XBOUND evaluates models at the state level and examines their ability to interpret multiple interface components and to unify semantically diverse instructions within the same screen. The framework defines two key tasks, Multi Widget Action Matching (MWAM) and Uni Widget Instruction Unification (UWIU), and introduces an Exploration Metric to quantify how well an agent understands a given state. Experiments on several existing GUI agents demonstrate that XBOUND reveals detailed capability boundaries, identifying strengths in action grounding and weaknesses in semantic generalization. The main contribution of this work lies in its evaluation perspective, providing a diagnostic view of model competence rather than direct performance improvement.

**Strengths:**

1. The paper introduces a new evaluation paradigm that shifts the focus from instruction-level accuracy to state-level understanding, providing a more holistic assessment of GUI agents’ real capabilities.
2. The proposed XBOUND framework, with its two complementary tasks (MWAM and UWIU) and the Exploration Metric, offers a systematic and interpretable way to analyze model performance across different capability dimensions.
3. The authors conduct extensive experiments on a diverse set of device-control agents, yielding insightful analyses that highlight specific strengths and weaknesses of existing models, which can guide future research and model design.

**Weaknesses:**

1. The XBOUND framework shows limited scalability and representativeness when applied to complex applications such as shopping or map apps. The evaluation only identifies visible and clickable UI elements but does not specify how coverage or selection is handled when a screen contains many interactive components, leaving representativeness unclear. Moreover, the framework lacks mechanisms for handling multi-step interaction flows and focuses solely on static states, which may lead to underestimation or misjudgment of model performance in realistic full-interface deployments, especially for complex real-world apps.
2. The dataset construction relies heavily on GPT-generated instructions and action pairs. However, the paper does not provide a quantitative assessment of annotation accuracy or inter-rater validation. Without human verification, it is unclear how reliable the generated data and task labels are, especially in cases of ambiguous UI semantics or icons.
3. XBOUND is only evaluated on Android-based GUI datasets. It remains unclear whether the framework generalizes to other environments, such as web interfaces or desktop applications, which limits its applicability to broader GUI agent research.

**Questions:**

1. How does XBOUND handle states with a very large number of interactive components? Are all clickable widgets included, or is there a sampling or filtering strategy? If so, what specific sampling or filtering strategy is adopted?
2. Could the authors formalize or operationalize how “capability boundaries” are quantitatively identified from the state-level metrics? Specifically, does the notion of capability boundaries encompass multiple dimensions such as visual perception, action grounding, and interaction reasoning, and if so, how are these dimensions separately measured or combined into a unified metric?
3. The paper claims that grounding data improves MWAM performance, while trajectory data benefits UWIU tasks, but the mechanism behind this distinction is unclear. Could the authors elaborate on why these two data types affect different evaluation scenarios differently?
4. How would the authors expect XBOUND to generalize to web or desktop interfaces where interaction patterns differ significantly?

---

### Official Review · Reviewer_uc45 · 2025-10-28

**Soundness:** 3
**Presentation:** 2
**Contribution:** 2
**Rating:** 4
**Confidence:** 3

**Summary:**

1. This paper presents XBOUND, a state-level evaluation framework for device-control agents. It argues that current evaluation methods at the instruction level may overlook the complexity of GUI states that contain multiple widgets and diverse instructions.

2. XBOUND defines two scenarios: Multi-Widget Action Matching (MWAM) to test distinguishing among multiple interactive widgets, and Uni-Widget Instruction Unification (UWIU) to test unifying diverse instructions for the same widget. An Exploration Metric is introduced to quantify the extent of state mastery.

3. The authors evaluate 11 open-source agents in the mobile use domain. Results show that UI-TARS, the strongest 7B model, achieves the best overall performance, while sub-7B models remain limited in mastering states. A bimodal performance pattern is observed in instruction unification.

4. The study identifies GPT-based planning as a critical bottleneck. It also shows that grounding data primarily improves action matching, whereas trajectory data is more effective for instruction unification, and highlights several challenging states such as widget disambiguation and dynamic state understanding.

**Strengths:**

1. This paper identifies an important limitation of current evaluation methods that operate only at the instruction level and motivates the need for state-level analysis in device-control agents.

2. It introduces two concrete evaluation scenarios, Multi-Widget Action Matching and Uni-Widget Instruction Unification, which capture distinct challenges faced by agents in realistic GUI environments.

3. The Exploration Metric provides an intuitive and interpretable way to quantify the extent of state mastery, offering a finer-grained perspective beyond overall task success.

4. The experiments section covers 11 open-source agents in the mobile use domain and reveals several insights, including the superiority of UI-TARS at the 7B scale, the bimodal performance pattern in instruction unification, and the role of GPT-based planning as a bottleneck.

**Weaknesses:**

1. The novelty of XBOUND lies mainly in shifting the evaluation perspective to the state level. This is useful, but the conceptual advance over existing evaluation methods could be articulated more clearly.

2. The experiments section covers 11 DC agents, yet the choice of only two scenarios may not fully capture the variety of challenges DC agents face in complex GUI environments.

3. GPT-based planning is highlighted as a bottleneck, but the discussion of why it arises and how general this limitation is remains somewhat limited.

4. The dataset construction is important for XBOUND, but details on validation and reliability checks are not fully clear, which may affect confidence in the results.

**Questions:**

1. Could the authors further clarify what unique insights state-level evaluation provides compared to instruction-level or task-level evaluation?

2. This paper chooses Multi-Widget Action Matching and Uni-Widget Instruction Unification as the two evaluation scenarios. Could the authors elaborate on why these two are representative and whether other scenarios might also be important?

3. The results highlight GPT-based planning as a critical bottleneck. Could the authors discuss whether this limitation is fundamental to current DC agents, or more specific to the setup of XBOUND?

4. The dataset construction plays a central role in supporting XBOUND. Could the authors provide more details on how they ensured the consistency and robustness of the evaluation data?

---

### Official Review · Reviewer_6bAH · 2025-10-29

**Soundness:** 2
**Presentation:** 2
**Contribution:** 2
**Rating:** 2
**Confidence:** 3

**Summary:**

The paper proposes XBOUND, a state-level evaluation method for GUI-based device-control agents.
Instead of measuring accuracy per instruction, XBOUND aggregates performance across all instructions within each screen (“state”), defining two scenarios:
1. Multi-Widget Action Matching (MWAM): distinguishing different widgets under the same state.
2. Uni-Widget Instruction Unification (UWIU): mapping diverse instructions to the same widget.
Using an augmented AndroidControl dataset, the authors evaluate 11 open-source agents (<7B).
They find UI-TARS performs best, grounding data helps MWAM, and trajectory data helps UWIU.

**Strengths:**

1. Highlights a useful state-level evaluation perspective beyond instruction-level metrics.

2. Defines two clear capability types (MWAM/UWIU) with interpretable results.

3. Provides broad empirical comparison across 11 agents and multiple tasks.

**Weaknesses:**

1. Method is simple — essentially averaging per-state accuracy, not a fundamentally new metric.

2. Data quality concerns: large portions of the dataset are LLM-generated without clear validation.

3. Limited scope: experiments cover only Android mobile GUIs.

4. No link to real performance: unclear if state-level scores predict full-task success.

5. Claims overstated: conclusions about bottlenecks or superiority lack ablation or statistical support.

**Questions:**

1. Does higher state-level EM correlate with full-task success?

2. Can XBOUND generalize beyond Android (e.g., web or desktop)?

3. Can the authors provide a quantitative breakdown of the main failure types observed in challenging states?
For example, how often do issues such as icon misrecognition, similar-icon confusion, incorrect action order, or poor environment awareness occur among all low-performance states?

---

### Official Review · Reviewer_3kPw · 2025-10-30

**Soundness:** 2
**Presentation:** 2
**Contribution:** 2
**Rating:** 2
**Confidence:** 3

**Summary:**

This paper introduces XBOUND, a state-level evaluation metric for device-control agents (DCAs), with the goal of providing a more granular analysis than standard instruction-level evaluation. The concept of assessing an agent's proficiency within individual GUI states is novel and addresses a relevant need in the field for finer-grained diagnostic tools.

**Strengths:**

The paper tackles an important and timely problem in DCA evaluation. The shift in focus from binary task success/failure to a nuanced, state-centric analysis is a commendable and necessary direction for the community. The conceptual division of agent capabilities into Multi-Widget Action Matching (MWAM) and Uni-Widget Instruction Unification (UWIU) is well-founded, as it captures two distinct and essential skills for robust GUI interaction. The experimental work is extensive, involving a systematic evaluation of 11 different agents, and some empirical observations, such as the bimodal performance distribution in UWIU and the distinct roles of grounding versus trajectory data, provide valuable, practical insights for researchers and developers.

**Weaknesses:**

The paper does not adequately justify the advantage of its proposed Exploration Metric (EM) over established instruction-level or task-level success rates. It claims that existing methods overlook broader contextual interactions but fails to provide a clear, quantitative, or qualitative comparison showing how XBOUND offers a superior or more informative assessment. The explanation of the metric itself, particularly the supporting figure, is confusing and difficult to understand, hindering the reader's ability to grasp its novelty and operational mechanics.

Furthermore, for a contribution centered on a new evaluation metric, a fundamental omission is the validation of XBOUND against existing standards. The authors do not investigate the correlation between an agent's performance on the state-level EM and its end-to-end task-level success rate. This lack of validation leaves a critical question unanswered: does high performance on XBOUND actually predict success in real-world, multi-step tasks? Without establishing this link, the practical relevance and predictive power of the metric remain unproven.

The evaluation also insufficiently addresses the relationship between state-level performance and overall task performance. The evaluation is conducted in a state-isolated manner, devoid of trajectory history. While this isolation is a design choice, the authors must discuss the implications of this approach and how, or if, proficiency in isolated states composes into successful task completion. The heavy reliance on LLMs for generating the "pseudo" trajectory tree dataset introduces another significant weakness, as potential biases or errors from the model-generated instructions and actions could compromise the integrity of the evaluation benchmarks without a thorough analysis of their impact.

**Questions:**

1. Could the authors provide a clear, concrete example where standard evaluation fails and XBOUND offers a superior diagnostic perspective?
2. Could the authors further explain (or analyze quantitively) the correlation between the state-level Exploration Metric and end-to-end task success rates to validate XBOUND's relevance?

---

### Official Review · Reviewer_7J7n · 2025-10-31

**Soundness:** 2
**Presentation:** 2
**Contribution:** 1
**Rating:** 2
**Confidence:** 3

**Summary:**

XBOUND is an evaluation paper. It argues that today’s GUI agents are usually tested one-instruction at a time and miss what really happens inside a UI “state” that contains many clickable widgets. The paper proposes a state-level evaluation with two scenarios—Multi-Widget Action Matching (can the agent pick the right widget for different instructions on the same screen?) and Uni-Widget Instruction Unification (can the agent map different phrasings to the same correct action?). It quantifies them with an Exploration Metric. They test 11 sub-7B models on mobile tasks, the paper reports: UI-TARS is the best 7B family; many models show bimodal behavior on unification (either very good or very bad); grounding data mostly helps action matching while trajectory data helps instruction unification. The paper also lists “hard states” such as ambiguous icons, similar-looking widgets, action order (topology), and poor state awareness.

**Strengths:**

1. Clear reframing of evaluation. Shifting to state-level thinking is useful. It better reflects how real agentic situations present multiple, competing targets and how one state can support many valid instructions. This framing makes failure modes easier to spot (e.g., “keeps clicking the search box even when the camera icon is the goal”).
2. Action Matching vs Instruction Unification is simple and useful distinction -- even if not worded in simple terms. It distinguishes perception errors (can the agent find the right widget?) from semantic/decision errors (can the agent realize different wordings mean the same action). That separation is helpful for diagnosis and dataset design.
3. The paper names four recurring issues—missing icon knowledge, similar icons, action order, and state misread. That gives future model designs actionable insights to look for.

**Weaknesses:**

1. The paper does not offer training recipes for reliable direct control (e.g. ways to recover from off-policy drift seems to an important challenge here).
2. While many systems are compared, the discussion rarely digs into why a given architecture fails a given state beyond labeling it “planning” vs “grounding.” There’s little ablation on perception modules, planners, memory, or search -- it is unclear where the bottlenecks are, for example with perfect perception of icon knowledge, would the agents start performing well?
3. The Exploration Metric provides no guidance on how to fix the issues it enlightens (e.g., curriculum design, counterfactual replays, negative mining, temporal consistency losses could be potential solutions).
4. The paper points out GPT-based planning as a failure source but does not offer any solution.

**Questions:**

Please clarify weaknesses.

---

### Note · Authors · 2025-12-02

I have read and agree with the venue's withdrawal policy on behalf of myself and my co-authors.